# Pharmacogenetic Influences on Individual Responses to Ocular Hypotensive Agents in Glaucoma Patients

**DOI:** 10.3390/pharmaceutics17030325

**Published:** 2025-03-02

**Authors:** Sara Labay-Tejado, Virginia Fortuna, Néstor Ventura-Abreu, Mar Hernaez, Valeria Opazo-Toro, Alba Garcia-Humanes, Mercè Brunet, Elena Milla

**Affiliations:** 1Department of Ophthalmology (ICOF), Hospital Clínic de Barcelona, Universitat de Barcelona, Carrer de Sabino Arana 1, 08028 Barcelona, Spain; sclabay@clinic.cat; 2Department of Biochemistry and Molecular Genetics (CDB), Hospital Clínic de Barcelona, Carrer de Villaroel 170, 08036 Barcelona, Spain; vfortuna@clinic.cat (V.F.); or mhernace34@alumnes.ub.edu (M.H.); agarciah@clinic.cat (A.G.-H.); mbrunet@clinic.cat (M.B.); 3Department of Glaucoma (ICOF), Hospital Clínic de Barcelona, Universitat de Barcelona, Carrer de Sabino Arana 1, 08028 Barcelona, Spain; neventura@clinic.cat; 4Doctoral Program in Biomedicine, University of Barcelona, 08007 Barcelona, Spain; 5Skövde’s Ophthalmology Department, Skaraborg’s Hospital, Lövänsvägen, 549 49 Skövde, Sweden; 6Institut d’Investigacions Biomèdiques August Pi i Sunyer (IDIBAPS), Centro de Investigación Biomédica en Red de Enfermedades Hepáticas y Digestivas (CIBEREHD), Carrer del Rosselló 149, 08036 Barcelona, Spain

**Keywords:** glaucoma, intraocular pressure, visual field, pharmacogenetics, personalized treatment, single-nucleotide polymorphism, *PTGFR*, *CYP2D6*, *ADRB2*

## Abstract

**Background/Objectives**: To analyze the genotype that predicts the phenotypic characteristics of a cohort of patients with glaucoma and ocular hypertension (OHT) and explore their influence on the response to ocular hypotensive treatment. **Methods**: This was a prospective study that included 193 eyes of 109 patients with glaucoma or OHT under monotherapy with beta-blockers, prostaglandin, or prostamide analogues (BBs, PGAs, PDs). Eight single-nucleotide polymorphisms were genotyped using real-time PCR assays: prostaglandin-F2α receptor (*PTGFR*) (rs3766355, rs3753380); beta-2-adrenergic receptor (*ADRB2*) (rs1042714); and cytochrome P450 2D6 (*CYP2D6*) (**2* rs16947; **35* rs769258; **4* rs3892097; **9* rs5030656, and **41* rs28371725). The main variables studied were baseline (bIOP), treated (tIOP), and rate of variation in intraocular pressure (vIOP), and mean deviation of the visual field (MD). The metabolizer phenotype and the *CYP2D6* copy number variation were also evaluated. **Results**: In total, 112 eyes were treated with PGAs (58.0%), 59 with BBs (30.6%), and 22 with PDs (11.4%). For *PTGFR* (rs3753380), statistically significant differences were observed in vIOP in the PGA group (*p* = 0.032). Differences were also observed for *ADRB2* (rs1042714) in MD (*p* < 0.001) and vIOP (*p* = 0.017). For *CYP2D6*, ultrarapid metabolizers exhibited higher tIOP (*p* = 0.010) and lower vIOP (*p* = 0.046) compared to the intermediate and poor metabolizers of the BB group. Additionally, systemic treatment metabolized by *CYP2D6* showed a significant influence on vIOP (*p* = 0.019) in this group. **Conclusions**: These preliminary findings suggest the future potential of pharmacogenetic-based treatments in glaucoma to achieve personalized treatment for each patient, and thus optimal clinical management.

## 1. Introduction

Glaucoma is a blinding disease, and in younger patients its progression can be particularly devastating without timely and effective treatment. Therapeutic trials can be frustrating and time-consuming, often resulting in a prolonged period where ineffective eye drops fail to reach target IOP. During this time, patients may face repeated appointments and multiple prescriptions over months or even years.

Glaucoma is characterized by a progressive degeneration of retinal ganglion cells and their axons, resulting in subsequent visual impairment. Glaucoma is the leading cause of irreversible blindness; its global prevalence is projected to rise by 74%, reaching 111.8 million individuals by the year 2040 [1]. The risk factors in primary open-angle glaucoma (POAG), the most common subtype of glaucoma, include age, race, family history, and intraocular pressure (IOP), among others [2]. Despite being a multifactorial disease, IOP control remains the only known modifiable factor to prevent both structural and functional glaucomatous progression [3,4,5,6,7].

The most common drugs used to treat glaucoma are prostaglandin analogues (PGAs) and beta-blockers (BBs), with the former being preferred due to their superior efficacy and safety profile, and once-daily dosage regimens [8,9]. The rate of response can vary across populations, but a favorable response is typically standardized at a 25–35% and 20–25% IOP decrease for PGAs and BBs, respectively [10].

The pharmacological category of prostaglandin analogues has two chemically different subtypes of molecules: prostaglandin analogues (PGAs) (latanoprost, travoprost, tafluprost) and prostamides (PDs) (bimatoprost) [11]. Prostaglandins are pro-inflammatory molecules derived from the arachidonic acid metabolism by cyclooxygenase enzymes (COX). These acidic molecules interact with G-protein-coupled receptors expressed in the corneal and ciliary epithelium, ciliary muscle, iris stroma, and smooth muscle to exert their action [12,13]. Reductions in IOP have been linked to both short-term and long-term effects on uveoscleral outflow. Short-term effects in monkeys are attributed to smooth muscle relaxation, and long-term effects seem to include the expansion of intermuscular spaces through matrix metalloproteinase remodeling [8,12,13]. Latanoprost (the most used PGA) is a prodrug of prostaglandin F2 alpha, which upon crossing the cornea, undergoes hydrolysis by esterases, becoming biologically active and able to bind to the *prostaglandin F2 alpha receptor* (*PTGFR*) [14]. Moreover, it is the first drug to demonstrate preservation of the visual field (VF) in patients with POAG in a randomized placebo-controlled trial [15]. Nevertheless, some reports indicate that between 4.1% and 51.5% of patients obtain an IOP reduction of 15% or less [16]. Several studies have shown that switching to bimatoprost or travoprost can further reduce IOP in latanoprost non-responders [9,17,18], and others have revealed more hyperemia with bimatoprost in comparison to latanoprost [19,20,21,22].

In contrast to PGAs, PDs are prodrugs derived from the oxygenation of principal endocannabinoids by COX-2. They lack the carboxylic acid group, rendering them neutral in solution [23]. PDs’ mechanism of action remains controversial, with some researchers hypothesizing that they activate a specific “prostamide” receptor that has yet to be cloned [24].

BBs are widely used medications in the field of medicine. When topically administered, they are absorbed through the conjunctiva, but also the lacrimal channels, nasal mucosa, and gastrointestinal tract into the general circulation, with subsequent possible secondary effects on the respiratory (bronchospasm, respiratory failure) and cardiovascular systems (bradycardia, hypotension, heart failure). Their mechanism of action is the inhibition of aqueous humor production in the ciliary epithelium. *Adrenergic receptors* (*ADRB*) are also members of the large superfamily of G-protein-coupled receptors and have been historically divided into *ß1* and *ß2* subtypes, based on their affinity to catecholamines [25]. Therefore, among BBs we have non-selective (carteolol, timolol) and cardioselective *ß1*-adrenoreceptor-blocking drugs (betaxolol) (14). *ADRB1* and *ADRB2* receptors have been specifically identified in the trabecular meshwork, ciliary body, and optic nerve head [25].

Several factors, such as age, gender, pharmacogenetics, epipharmacogenetics, physiopathology, and adherence may explain the inter-individual diversity in therapeutic drug response. Focusing on pharmacogenetics, there are some types of gene variations (polymorphisms) that can influence the expression and activity of some metabolic enzymes or specific receptors that can determine the response to pharmacologic therapy (pharmacokinetics, efficacy, and toxicity) in glaucoma. Single-nucleotide polymorphisms (SNPs) are the most common DNA polymorphisms [13,26].

Some of them have been associated with a different response to latanoprost, namely rs3766355 and rs3753380. In particular, these two SNPs have been found to cause an alteration in *PTGFR* expression leading to a suboptimal response to treatment in terms of IOP reduction. More precisely, homozygous (HM) or heterozygous (HT) patients exhibited a smaller reduction in IOP compared to those carrying the homozygous “wildtype” (WT) allele [14,27,28,29,30,31]. Yet, some studies have indicated that there is no correlation between these variants and the ocular response to latanoprost [32].

In the case of BBs, the Arg389Gly polymorphism in *ADRB1* (rs1801253) has been associated with a higher baseline IOP (bIOP) and a greater reduction in IOP following topical betaxolol treatment [25]. In contrast, the Gln27Glu polymorphism in *ADRB2* (rs1042714) seems to decrease IOP by at least 20% in HM patients receiving treatment compared to HT patients. A similar trend is observed for WT patients, although it does not reach statistical significance [33].

In general, BBs are metabolized by the highly polymorphic enzyme *cytochrome P450 2D6* (*CYP2D6*), which also metabolizes approximately 25% of the systemic medication currently in use, such as antipsychotics, antidepressants, and antiarrhythmics [34]. This gene is located in chromosome 22 q13.2, and more than 100 associated polymorphisms have been reported [25]. *CYP2D6* exhibits four major phenotypes, with different frequencies in the global population: poor metabolizers (PMs; 0.4–5.4%), intermediate metabolizers (IMs; 0.4–11%), normal or extensive metabolizers (EMs; 67–90%), and ultrarapid metabolizers (UMs; 1–21%) [34,35]. Patients with two functional copies of the *CYP2D6* gene (*CYP2D6*1/*1* homozygotes or *CYP2D6*1/*2* heterozygotes) are referred to as EMs. PMs have two copies of a null allele. IMs have at least one copy of an allele with a limited enzymatic activity. In contrast, UMs are often associated with gene duplication (*CYP2D6*2XN*). However, genotyping for duplicated *CYP2D6* alleles only explains a fraction of 30–40% of the UM phenotypes observed in Caucasian populations. Enhancer SNP rs5758550 (H1a) could be a main source for the UM phenotype [25,34].

Clinically, PMs seem to have higher circulating timolol levels after treatment with topical timolol, thereby increasing the risk of bradycardia. UMs have been shown to have lower plasma concentrations of metoprolol than the other groups [25]. Studies have observed that homozygotes and heterozygotes for the SNP rs16947 (*CYP2D6 *2*) may increase the risk of developing bradycardia induced by timolol, and that the SNP rs769258 (*CYP2D6 *35* duplication-negative) is notably more prevalent in Caucasian UMs [34,36,37,38].

Notably, interest in the clinical implementation of pharmacogenetics has surged significantly over the past decade. This discipline aims to facilitate the selection of the appropriate drug and dose for each patient, thereby enhancing clinical outcomes. The understanding of genetic factors that contribute to variable drug response has also greatly expanded. As a result, there is now substantial evidence supporting the clinical implementation of pre-emptive pharmacological tests to provide recommendations and guide treatment across different therapeutic disciplines. Expert working groups have published guidelines for genotype-based drug and initial dose selection for certain medications to achieve a personalized treatment for each patient, e.g., the Clinical Pharmacogenetics Implementation Consortium (CPIC) and the Dutch Pharmacogenetics Working Group (DPWG), among others [14,39,40]. There is still not much research on pharmacogenetics and personalized medicine in glaucoma, but the area will soon benefit from these advancements.

This study aimed to perform pharmacogenetic testing on a cohort of patients with glaucoma or ocular hypertension (OHT) who had been treated with either PGAs, BBs, or PDs via monotherapy, and evaluate the association between the pharmacogenetic results (genotype) and the clinical response in terms of IOP control and VF severity. This represents a step forward compared to a prior publication by the authors, as it entails a study with an increased sample size, focusing exclusively on monotherapy and analyzing a broader array of parameters [14].

## 2. Materials and Methods

### 2.1. Participants and Sample Selection

A prospective, observational, and consecutive clinical case study was conducted on 109 patients affected by OHT or glaucoma undergoing monotherapy with topical PGAs (latanoprost 50 µg/mL, tafluprost 15 µg/mL, travoprost 40 µg/mL), BBs (carteolol 10 µg/mL, timolol 5 µg/mL, betaxolol 2.5 µg/mL), or PDs (bimatoprost 0.3 µg/mL). In the examined cohort, treatment was applied to 84 patients in both eyes, whereas 25 received it in one eye only. Consecutive patients were recruited from September 2020 to September 2023 in consultation rooms of the Ophthalmology Department of the Hospital Clínic of Barcelona. All patients underwent the complete ophthalmologic examination described below and blood samples were extracted for genotype analysis at the Pharmacology and Toxicology Unit of Biochemistry and Molecular Genetics of the same hospital. The investigation adhered to the principles outlined in the Declaration of Helsinki. Approval for the research protocol was granted by the ethics committee, and informed consent was obtained from every participant prior to their inclusion in the study (HCB/2019/1144). Regarding the diagnosis of glaucoma and OHT, the guidelines of the European Glaucoma Society (EGS) were followed. For the classification of glaucoma subtypes, the ICD-10 Glaucoma Reference Guide was adhered to.

### 2.2. Inclusion Criteria

We included patients aged 18 and above with OHT or any subtype of glaucoma (except those specified in the exclusion criteria paragraph) that were under a strict monotherapy regimen either with PGAs, BBs or PDs, irrespective of the dose and duration of the treatment, and who expressed interest in participating and were willing to provide a blood sample. IOP was measured at baseline, prior to the initiation of any ocular hypotensive medication, or after a minimum washout period of 3 months. Following this, IOP was re-measured no earlier than 3 months after initiating treatment with the studied medication, as some patients may exhibit a delayed response. Therefore, all patients were treated for a minimum of 3 months. The treatment regimen followed the appropriate dosing schedule depending on the type of medication: PGAs and PDs were administered once daily at night before bedtime, while BBs were administered twice daily. The selection of the appropriate treatment for each patient was personalized in accordance with the recommendations of the European Glaucoma Society (EGS), and was performed by a senior consultant (EM).

### 2.3. Exclusion Criteria

Exclusion criteria included uveitic, neovascular, traumatic, or other types of inflammatory glaucoma that required additional ocular medication; eyes that were in a phthisis bulbi condition; and patients that had undergone intraocular surgery within the past 3 months or were dealing with an ocular acute condition requiring any extra medication.

### 2.4. Data Collection

Enrolled participants underwent thorough anamnesis to collect data on their current and past eye and systemic medications and concomitant pathologies, a complete ophthalmological examination (best-corrected visual acuity, Goldmann applanation tonometry, slit lamp examination, ocular fundoscopy), and ancillary tests, namely ultrasound pachymetry, gonioscopy, Humphrey VF perimetry Analyzer 3 (HFA3, Carl Zeiss Meditec Inc., Dublin, CA, USA), and Cirrus optical coherence tomography (Carl Zeiss Meditec Inc., Dublin, CA, USA), in order to collect all the relevant data for the study. Regarding IOP measurements, all of them were recorded in the morning (during consultation hours) by two trained clinicians (SL and EM). The mean value of three readings, recorded at 8:00 AM, 11:00 AM, and 2:00 PM, was used for the analysis. The tonometer was regularly calibrated following the manufacturer’s guidelines and no adjustments were made based on the pachymetry result. Eight SNPs were genotyped using real-time PCR assays: *prostaglandin-F2α receptor* (rs3766355, rs3753380); *beta-2-adrenergic receptor* (rs1042714); and *cytochrome P450 2D6* (**2* rs16947; **35* rs769258; **4* rs3892097; **9* rs5030656, and **41* rs28371725). All of them were prior-selected based on their known associations with treatment response, even though the information available in the literature was limited.

### 2.5. Analysis of Genotypes

DNA was isolated and purified from the peripheral whole blood of the patients included in the study using MagNA Pure Magnetic Glass Particle (MGP) Technology. For genotyping *PTGFR*, *ADRB2*, and *CYP2D6* polymorphisms, an allelic discrimination reaction was performed using a specific Taqman Custom plating SNP from Applied biosystems (reference 4462782) with lyophilized probes and primers with the following polymorphisms: *PTGFR* rs3753380 (C___1686003_10), *PTGFR* rs3766355 (C__25807762_10), *ADRB2* rs1042714 (C___2084765_20), *CYP2D6 *2* rs16947 (C__27102425_10), *CYP2D6*35* rs769258 (C__27102444_F0), *CYP2D6*4* rs3892097 (C__27102431_D0), *CYP2D6*9* rs5030656 (C__32407229_60), and *CYP2D6*41* rs28371725 (C__34816116_20), using a QuantStudio 5 Real-Time PCR System (Thermo Fisher Scientific, Barcelona, Spain). An amount of 30 µL of DNA at 4 ng/µL from each sample or control is mixed with 30 µL of TaqPath™ ProAMP™ Master Mix (Applied Biosystems with ref. A30865) in an Eppendorf tube (a total of 60 µL for each sample or control). Then, 10 µL of each sample or control is dosed to the 5 wells of each SNP of the personalized plate, leaving a final concentration of 2 ng/µL. The plate is covered with adhesive (optical film). A centrifuge pulse is given (8″–10″ at 1800 rpm). Finally, the plate is analyzed in the QuantStudio 5 Real-Time PCR System. The thermal cycler conditions are summarized in Appendix A.

In addition, all DNA samples from patients treated with BBs were screened for *CYP2D6*_cnv (exon 9 Hs00010001 cn, intron 2 Hs04083572_cn, and intron 6 Hs04502391_cn) copy number variations (CNVs) using RT-PCR. DNA from each subject was tested for *CYP2D6* triplicates using TaqMan Copy Number Assays (Life Technologies, CA, USA). A segment of the gene was amplified in 3 replicates for each DNA sample. The experiment was conducted using a QuantStudio 5 Real-Time PCR System (Thermo Fisher Scientific, Barcelona, Spain), which quantitates the gene of interest, normalized to an endogenous reference gene (*RNase P*; ref.4403326 Applied biosystems, Life Technologies) known to be present in 2 copies of a diploid genome. An amount of 1 µL of DNA at 20 ng/µL from each sample or control is mixed with 19 µL of Mix (containing 10 μL of TaqMan™ Genotyping Master Mix ref.4371353 Applied Biosystems) in each well (MicroAMP Optical 96-well Plate ref. N8010560 Applied Biosystems), leaving a final concentration of 1 ng/µL. The plate is covered with a MicroAMP Optical 8-Cap Strip. A centrifuge pulse is given (8″–10″ at 1800 rpm). Finally, the plate is analyzed using the QuantStudio 5 Real-Time PCR System. The thermal cycler conditions are summarized in Table 1. The evaluation of the copy number of genomic DNA targets was performed using CopyCaller 2.1 software (Life Technologies) with the default settings.

### 2.6. Quality Analysis

The results were analyzed with Taqman Genotyper software (Software V.1.4) after determination using the QuantStudio 5 Real-Time PCR System (Thermo Fisher Scientific, Barcelona, Spain). The data obtained from the ophthalmological medical records and the data from the results of the PCR analysis were collected. We obtained the following data from the medical records: demographic features, bIOP, treatments used, tIOP, vIOP, medication side effects, diagnosis (angle-closure glaucoma, POAG, secondary glaucoma, or OHT), MD, VFI, PSD, previous ocular interventions, and systemic concomitant medication treatments or organic insufficiencies. For bIOP, we chose the value of IOP measured without any treatment or after the washout period if they had undergone any glaucoma surgery or were using any ocular hypotensive treatment. For this reason, patients who had undergone glaucoma surgery within the last three months (<3 months) were excluded to respect the washout period, as well as those with any acute ocular condition that required additional medication.

### 2.7. Data Analysis

IBM SPSS Statistics version 29 was used for the analysis of the data. Demographic information, including sex, age, race, eye laterality, and other data, such as type of glaucoma, eye surgery, topical ocular and systemic treatment, adverse drug effects, and presence of renal, cardiac, and liver insufficiencies, were summarized using frequency tables. Descriptive statistics such as mean, median, standard deviation, and range were calculated for these variables. The variants in each examined SNP (HM, HT, and WT), the metabolizer phenotype (PM, IM, EM, and UM), IOP parameters (bIOP, tIOP, and vIOP), and VF parameters (MD, VFI, and PSD) are also described in frequency tables. vIOP was calculated as the difference between tIOP and bIOP, dividing this value by bIOP and then multiplying by 100.

Quantitative data, specifically bIOP, tIOP, vIOP, MD, VFI, and PSD were assessed and compared to a theoretical normality curve using the Kolmogorov–Smirnov test with Shapiro–Wilks correction. The Kruskal–Wallis test was generally utilized for further analysis due to the significant deviation from normality. However, in the specific case of evaluating whether systemically metabolized medication by *CYP2D6* influences the response to topical BBs, the Mann–Whitney–Wilcoxon test was employed. An additional correlation analysis using pachymetry measures was performed. Statistical significance was determined for all results where the *p*-value was less than 0.05. Post hoc analyses of statistically significant results were conducted using the Bonferroni correction for multiple comparisons (see Appendix A).

Both eyes of each patient were included when available and analyzed separately, recognizing the asymmetry of glaucoma and the potential variation in topical hypotensive treatment between eyes.

A reduction in IOP of less than 25% for patients treated with PGAs or PDs, and less than 20% for those treated with BBs was considered a poor response. The mean percentage reductions in IOP were also compared.

## 3. Results

### 3.1. Clinical Features

Of the total of 109 patients recruited at the Ophthalmology Department of the Hospital Clínic of Barcelona during the inclusion period (2020–2023), the vast majority (93.58%) were of European descent and 57 (52.29%) were female. In total, 193 eyes were included, of which 97 (50.3%) were right eyes. The mean age was 66.10 ± 12.29 years, with 71.56% of the patients aged over 60 years (78 from 109). The main diagnosis was POAG (56%), and the second one was OHT (29.5%). Out of the 193 eyes, 117 (60.6%) had not undergone any ocular surgery, 34 (17.6%) eyes had previously required glaucoma surgery, and 36 (18.7%) had undergone other kinds of ophthalmological surgery in the past as well. Additionally, 26 (13.5%) eyes had undergone laser procedures, including iridotomy, iridoplasty, and selective laser trabeculoplasty (SLT). In reference to medical management, 112 eyes (58%) were treated with PGAs, 59 (30.6%) with BBs, and 22 (11.4%) with PDs. Systemic side effects, such as bradycardia or arrhythmia, were reported in nine patients (8.26%), of whom five were also receiving systemic medication metabolized by *CYP2D6*. In all of these cases, a switch in the therapeutic regimen was performed. As baseline health issues, 12 patients suffered from renal insufficiency, 3 from cardiac insufficiency, and 1 from liver impairment. A detailed demographic description is shown in Table 1.

The mean and standard deviation values for baseline IOP (bIOP), treated IOP (tIOP), and rate of IOP variation (vIOP) were 23.57 ± 4.80 mmHg, 19.99 ± 4.87 mmHg, and −14.04 ± 23.06 mmHg, respectively. The mean and standard deviation for the MD value of VF were −4.83 ± 7.22 decibels (dB). More details are specified in Appendix A.

### 3.2. Genetic Features

The SNP frequencies of the *PTGFR* and *ADRB2* genes in the selected cohort are outlined in Table 2 (A and B), with the sample size (N) referring to the number of patients rather than eyes, as in the rest of the analysis.

In Table 2C, the frequencies of the different alleles of the *CYP2D6* gene can be found. Note that for alleles *4 (rs3892097), *41 (rs28371725), and *9 (rs5030656), the information obtained from the BB subgroup (N = 27) is extrapolated to the total study population (N = 109). Among the BB group, the most frequent metabolizer phenotype was EM, observed in 12 patients (44.44%), followed by IM in 11 patients (40.74%), UM in 3 patients (11.11%), and PM in 1 patient (3.70%). These results are related to the CNVs, as among the UM patients, two had a CNV of 3 and one had a CNV of 4 (duplication-positive). The PM patient presented a CNV of 1, indicating a deletion.

Refer to Appendix A for a detailed description of the frequencies of the eight *CYP2D6* SNPs among the BB group, including their CNVs and the resulting metabolizer phenotypes.

All studied SNPs, except *ADRB2* rs1042714, were found to be in Hardy–Weinberg equilibrium.

Regarding the SNP rs3766355 of the *PTGFR* gene, no statistically significant differences in IOP and VF parameters were observed (Table 3). However, the trends were consistent in both: higher vIOP was found in WT (−17.23 ± 16.44 mmHg) compared to HT −8.10 ± 24.80 mmHg) and HM (−11.64 ± 24.43 mmHg) and better values of MD were found in WT (−2.49 ± 1.91 dB) in contrast to HT (−3.49 ± 6.13 dB) and HM (−4.45 ± 5.99 dB) in the PGA group (Figure 1 and Figure 2).

Concerning the SNP rs3753380 in the *PTGFR* gene, statistically significant differences in terms of vIOP were detected among the PGA group: WT = −15.94 ± 22.10 mmHg and HT = −2.99 ± 25.23 mmHg with *p* = 0.032 (Figure 3). The results for the VF parameters were not remarkable in this group (Table 4).

Regarding the PD group, no conclusive results were found in either VF or IOP for the SNPs rs3766355 or rs3753380 in the *PTFGR* gene. However, see Appendix A for more detailed information.

For the *ADRB2* gene BB group, no statistically significant differences in the VF or IOP parameters were observed (Table 5). However, a trend was noted: WT (−1.13 ± 1.53 dB) exhibited better MD values than HM (−5.35 ± 9.92 dB) and HT (−6.93 ± 6.78 dB), *p* = 0.064 (Figure 4). Additionally, higher vIOP was observed in WT (−27.00 ± 15.89 mmHg) and HM (−20.93 ± 21.09) compared to HT (−15.70 ± 24.52 mmHg) in this group (Figure 5).

When analyzing the whole cohort, statistically significant differences in the VF MD values of the SNP rs1042714 were observed, as WT (−1.22 ± 2.89 dB) presented better MD values than HT (−6.51 ± 7.41 dB) individuals, with *p* < 0.001. When comparing HM (−4.89 ± 7.72 dB) with HT (−6.51 ± 7.41 dB), the same trend was maintained, although it did not reach statistical significance (see Appendix A). These differences were also found for the rest of the VF parameters (VFI, PSD). Refer to Appendix A for IOP and VF parameters based on genetic profiles.

Regarding the metabolizer phenotypes (PM, IM, EM, and UM) derived from the eight *CYP2D6* SNPs analyzed in the BB group, a statistically significant difference was observed in both tIOP (*p* = 0.010) and vIOP (*p* = 0.046). Specifically, UM individuals exhibited higher tIOP (25.20 ± 3.70 mmHg) and lower vIOP (−10.76 ± 8.71 mmHg) compared to the EM (17.43 ± 4.74 mmHg; −26.80 ± 15.10 mmHg), IM (18.33 ± 4.51 mmHg; −13.12 ± 24.49 mmHg), and PM (14.00 ± 1.41 mmHg; −48.15 ± 5.24 mmHg) phenotypes, respectively (see Appendix A). The results for the VF parameters were not remarkable for *CYP2D6* SNPs. Below, Table 6 shows detailed information on this.

In addition, statistically significant differences in vIOP (*p* = 0.019) were found in the BB group between those patients taking systemic treatment metabolized by *CYP2D6* (−9.45 ± 22.09 mmHg) and those not taking it (−26.19 ± 18.87 mmHg) (Figure 6). No influence of renal (*p* = 0.917), cardiac (*p* = 0.414), or liver (*p* = 0.380) insufficiencies was detected on these vIOP values.

Finally, a weak but statistically significant positive correlation (0.168, *p* = 0.046, n = 141) was observed between corneal pachymetry and tIOP, especially in the PGA group (0.318, *p* = 0.002, n = 89). This correlation was not statistically significant in terms of vIOP (0.1000, *p* = 0.272, n = 123).

## 4. Discussion

Achieving a target pressure is one of the most important aspects of managing glaucomatous disease. Genetic variability, adherence to treatment, environmental factors, and both ocular and systemic diseases are believed to influence the overall IOP response to glaucoma medications [33]. It is known that a significant proportion of patients are clinical non-responders to hypotensive medication (termed “recalcitrant glaucoma”) [12]. The non-response rate differs among several studies and populations, ranging from 4.1% to 51.5% for PGAs, and is very variable for BBs, occasionally with an increased risk of systemic adverse effects [16,25]. Various SNPs in the *PTGFR*, *ADRB2,* and *CYP2D6* genes can affect the response to pharmacologic therapy in glaucoma [14].

This study aimed to perform pharmacogenetic testing on a cohort of patients with glaucoma and OHT who were undertaking solo treatment with either PGAs, BBs, or PDs, and correlate the results of their genetic profiles with their clinical response in terms of IOP control and VF severity. This project builds on the authors’ previous publication on pharmacogenetics in glaucoma by increasing the sample size, focusing on monotherapy, and examining a broader range of parameters [14]. Performing the study only on patients undertaking monotherapy may reduce the applicability to daily clinical practice, but it allows for a more precise and targeted identification of genetic effects on drug response.

Notably, the previous study by these authors is the only one to date to analyze and report an association of *PTGFR* rs3766355 with VF parameters (MD), observing significantly higher tIOP and more advanced MD values in HM and HT compared to WT individuals [14]. Although we did not find statistically significant differences in IOP or VF, the expected trends were observed in both for the PGA group. In contrast, other studies have focused exclusively on IOP. For instance, Cui et al. reported a significant association with genotype, whereas Zhang et al. initially observed no correlation on day 7, which became statistically significant by day 30 in Han Chinese patients [29,31]. Similarly, *Gao* and *Sakurai* et al. reported no association between bIOP and vIOP [27,30]. Consistent with these findings, we also did not observe an association between bIOP and this SNP genotype.

Regarding the SNP rs3753380 of the *PTGFR* gene, our study found that HT patients exhibited a statistically significant poorer response to PGAs compared to WT patients in terms of IOP variation (<25% IOP decrease). Unlike our study, the previous one reported differences exclusively in VF parameters [14]. Ussa et al. obtained similar findings regarding IOP control in their case–control study conducted in a Spanish population undergoing latanoprost monotherapy [28]. Sakurai et al. performed two studies on the SNPs rs3766355 and rs3753380 in Japanese volunteers. The first study found that both SNPs in the promoter and intron 1 regions of the *PTGFR* gene correlated with the response to latanoprost. However, the result of the haplotype analysis showed that one haplotype, which had a minor allele in rs3753380 and major ones in other SNPs, was significantly correlated with the response to this drug [27]. The second one found only rs3753380 to be significantly correlated with low responders through the downregulation of *PTFGR* [27]. However, most studies investigating the SNP rs3753380 did not find differences in the response [29,31]. The rs3766355 polymorphism appears to be the most common variant linked to a different response to PGAs, as previously reported [14,31]. However, McCarty et al. found no association between the SNPs rs3766355 and rs3753380 and PGAs in a European-descent American population, but without describing the specific drug studied [32]. The findings of this group may explain the smaller differences observed in our study, particularly for rs3766355, as the selected population is similar. This contrasts with other studies, which have primarily focused on Asian populations.

Regarding the SNP rs1042714 of the *ADRB2* gene, our study observed that HM subjects of the BB group were more likely to experience a significant IOP decrease and exhibit less severe glaucoma compared to HT subjects, although this difference did not reach statistical significance. Similar results were reported by McCarty et al., who observed a comparable trend in terms of IOP response. In contrast to our study, they did not find an association between the use of systemic BBs and IOP response [33]. Concretely, we observed that patients receiving systemic and topical BBs simultaneously presented a decreased IOP response, which might be explained by receptor saturation. The first population-based study on this topic seems to endorse these findings because it suggests that the IOP-lowering effect of topical BBs may be blunted in patients already using oral BBs, while increasing the probability of associated side effects [41]. The adverse effects in our study were scarce, so their relationship with the pharmacogenetic profile could not be evaluated with sufficient accuracy.

Regarding *CYP2D6*, our study found that the metabolizer phenotype significantly influences IOP response to topical BBs. *CYP2D6 *35* rs769258 is notably more prevalent in Caucasian UMs, according to a limited number of studies, leading to a rapid elimination of these drugs from the organism and potentially resulting in a hypo or a lack of response [36]. This aligns with our findings, which suggest that patients in the BB group classified as UMs exhibited a poorer response in terms of IOP reduction compared to the rest of the metabolizer groups (EMs, IMs, and PMs).

Regarding the results obtained on the potential weak influence of a thick pachymetry on a lower response to PGA treatment, no specific literature currently exists on this topic. The only related findings described in the literature pertain to the association between PGAs and increased corneal hysteresis, which is attributed to their direct effects on the viscoelastic properties of the cornea, irrespective of IOP reduction [42]. However, this could represent a contributing factor in the variability of response to these medications.

An essential point to consider is that, although some of our results may not be statistically significant, they can still have an important clinical impact. In fact, the Early Manifest Glaucoma Trial (EMGT) and the Ocular Hypertension Treatment Study (OHTS) have shown that lowering IOP by even 1 mm Hg is clinically relevant and reduces the risk of glaucoma progression or the development of glaucoma by approximately 10% [43]. In relation to the values obtained for visual field parameters, this is the second time in the literature that a possible association between the pharmacogenetic profile in glaucoma and visual field characteristics has been identified. This could be explained by the fact that a worse response to medication leads to increased disease progression, which translates into higher MD values. This aspect would be interesting to assess in future studies. Alternatively, certain SNP phenotypes may be directly associated with more severe VF defects.

Our study presents certain limitations. On one hand, the problem of non-adherence in glaucoma is a persistent issue that can affect the results [44]. However, as all patients included were on strict monotherapy regimens, compliance should have been much better than in patients receiving complex treatments with two or more agents, as suggested in the literature, although there are also studies that have found no such association [45]. Furthermore, IOP fluctuations may be another source of confusion, as IOP was only measured during office hours. Ideally, a tension curve may have reduced this bias, but this was unlikely to be achieved in daily clinical practice. Moreover, most of the patients were of European descent, and ethnicity seems to be an influential factor in treatment response. For example, two recent studies have indicated that timolol is less effective in the black population due to a mechanism that still remains unknown [25]. This is a highly underexplored topic, and as such, the exact factors involved in the pharmacogenetics of glaucoma are not yet fully understood. For this reason, in addition to performing genetic analyses based on treatment groups, we also conducted analyses across the entire cohort, irrespective of the treatment administered, to determine if there was any shift in trends in the results that could suggest a potential synergy or antagonism between the different drugs. However, the most notable limitation is the small sample size in certain subgroups. Without sufficient statistical power, differences may have existed without being detected due to this reason. Nevertheless, our study holds clinical relevance by advancing personalized medicine. In the near future, identifying these markers before treatment initiation will enhance drug selection for improved efficacy and reduced toxicity. Larger-scale glaucoma pharmacogenetic studies focused on monotherapy are necessary to draw more reliable conclusions related to the clinical and economic benefits of implementing anticipative pharmacogenetic tests in daily clinical practice.

## 5. Conclusions

Our results show that the pharmacogenetics of *CYP2D6* (**2* rs16947; **35* rs769258; **4* rs3892097; **9* rs5030656; and **41* rs28371725), *ADRB2* (rs1042714), and *PTGFR* (rs3766355 and rs3753380) have an impact on individuals’ responses to glaucoma treatment involving beta-blockers, prostaglandin analogues, or prostamides, respectively. *CYP2D6* ultrarapid metabolizers exhibit a poorer IOP response to BBs.

Further multicentric studies are required to confirm these preliminary findings in selected cohorts of glaucoma patients before integrating pre-emptive pharmacogenetic panels into routine clinical practice. This emerging approach holds promise for personalizing treatment, enabling a faster and more efficient selection of effective therapies while avoiding those likely to fail.

## Figures and Tables

**Figure 1 pharmaceutics-17-00325-f001:**
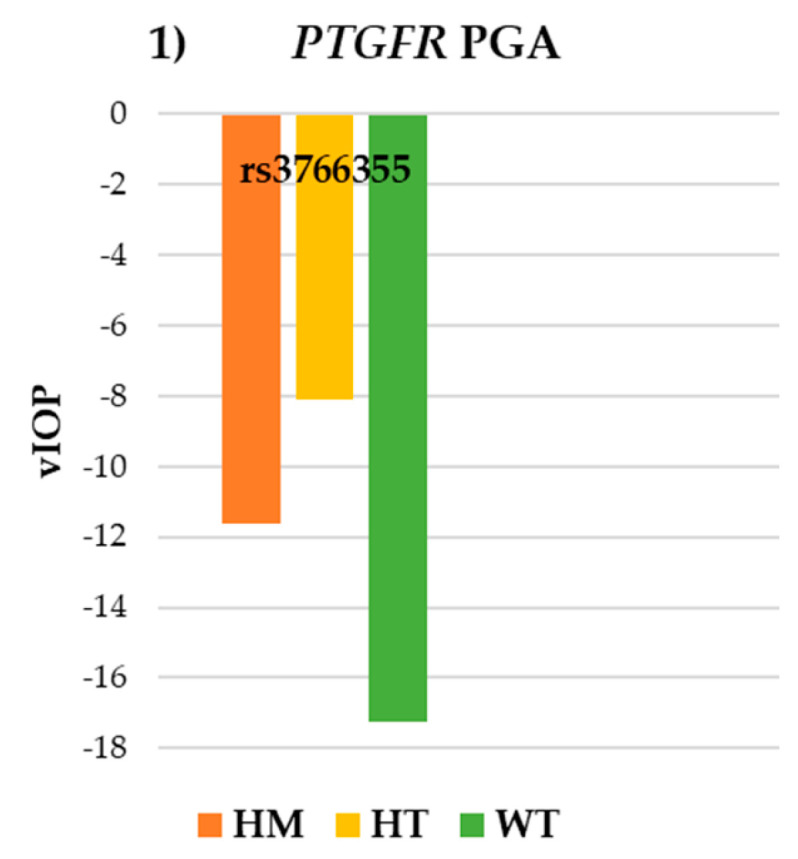
Intraocular pressure results for SNP rs3766355 of the *PTGFR* gene in the PGA group. *PTGFR*: prostaglandin-F2α receptor; PGA: prostaglandin analogues; WT: wildtype; HT: heterozygote; HM: homozygote; vIOP: rate of intraocular pressure variation.

**Figure 2 pharmaceutics-17-00325-f002:**
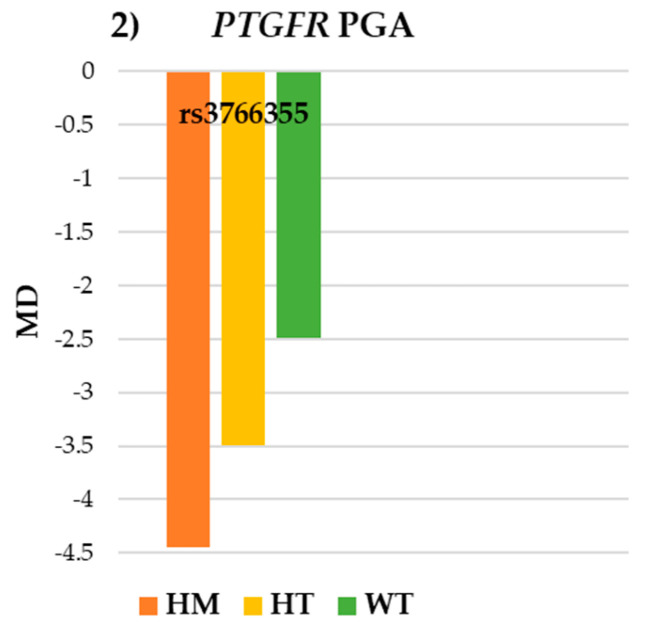
Visual field results for SNP rs3766355 of the *PTGFR* gene in the PGA group. *PTGFR*: prostaglandin-F2α receptor; PGA: prostaglandin analogues; WT: wildtype; HT: heterozygote; HM: homozygote; MD: medium deviation.

**Figure 3 pharmaceutics-17-00325-f003:**
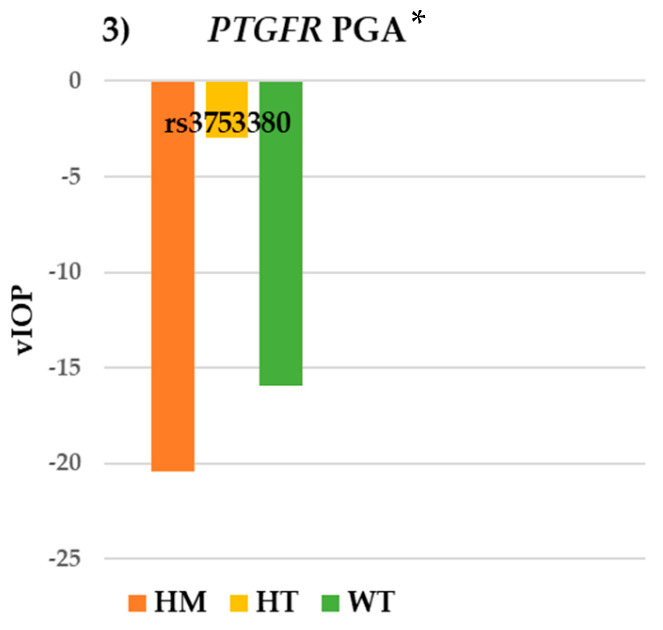
Intraocular pressure results for SNP rs3753380 of the *PTGFR* gene in the PGA group. *PTGFR*: prostaglandin-F2α receptor; PGA: prostaglandin analogues; WT: wildtype; HT: heterozygote; HM: homozygote; vIOP: rate of intraocular pressure variation. * *p* < 0.05 (indicating statistical significance).

**Figure 4 pharmaceutics-17-00325-f004:**
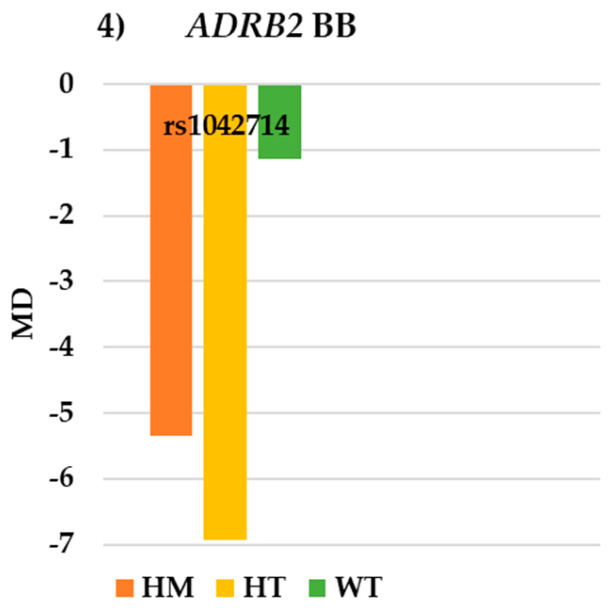
Visual field results for SNP rs1042714 of the *ADRB2* gene in the BB group. *ADRB2*: beta-2 adrenergic receptor; BB: beta-blockers; WT: wildtype; HT: heterozygote; HM: homozygote; MD: medium deviation.

**Figure 5 pharmaceutics-17-00325-f005:**
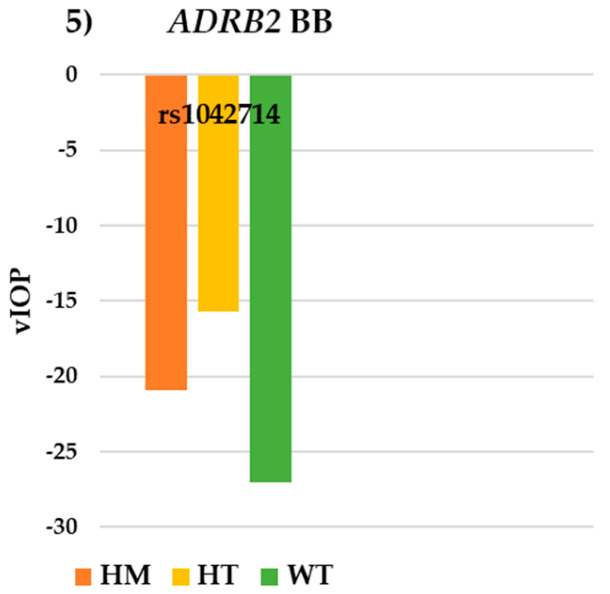
Intraocular pressure results for SNP rs1042714 of the *ADRB2* gene in the BB group. *ADRB2*: beta-2 adrenergic receptor; BB: beta-blockers; WT: wildtype; HT: heterozygote; HM: homozygote; vIOP: rate of intraocular pressure variation.

**Figure 6 pharmaceutics-17-00325-f006:**
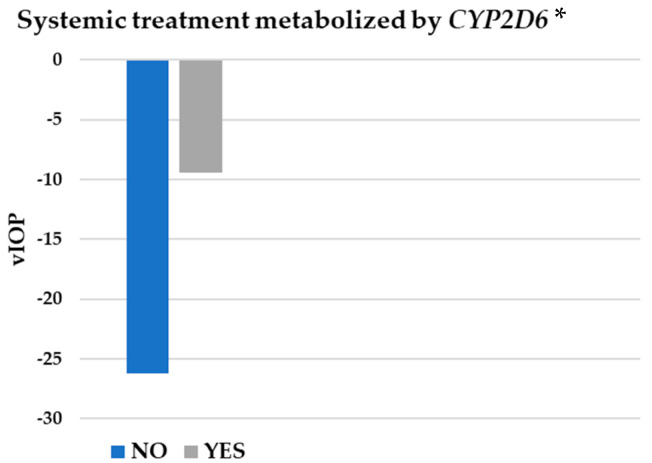
Influence of systemic treatment metabolized by *CYP2D6* on intraocular pressure. *CYP2D6*: cytochrome P450 2D6; vIOP: rate of intraocular pressure variation. * *p* < 0.05 (indicating statistical significance).

**Table 1 pharmaceutics-17-00325-t001:** Demographic and clinical features of the study cohort.

Features		Frequency	Percentage (%)
**Sex**	Female	57	52.29
	Male	52	47.71
**Race**	European descent	102	93.58
	South American	3	2.75
	Asian	4	1.83
**Eye laterality**	Right	97	50.3
	Left	96	49.7
**Eye diagnosis**	Primary open-angle glaucoma	108	56
	Angle-closure glaucoma	18	9.3
	Ocular hypertension	57	29.5
	Secondary glaucoma (pseudoexfoliative, pigmentary)	10	5.2
**Eye surgery records**	No previous ocular surgery	117	60.6
	Glaucoma surgery	34	17.6
	Filtering procedures (non-penetrating deep sclerectomy, trabeculectomy)	21	
	Devices (Paul, Baaerveldt)	4	
	MPEGS (XEN, Express)	5	
	MIGS (iStent^®^)	4	
	Laser	26	13.5
	Selective laser trabeculoplasty	6	
	Iridotomy	15	
	Iridoplasty	5	
	Other eye surgery (cataracts, refractive or retinal surgery)	36	18.7
**Hypotensive treatment per eye**	Prostaglandin analogues	112	58
	Latanoprost	98	
	Travoprost	2	
	Tafluprost	12	
	Beta-blockers	59	30.6
	Carteolol	48	
	Timolol	9	
	Betaxolol	2	
	Prostamides (bimatoprost)	22	11.4
**Systemic adverse effect**	No	100	91.74
	Yes	9	8.26
**Systemic treatment *CYP2D6***	No	57	52.29
	Yes	52	47.71
**Organ dysfunction**	Renal insufficiency	12	11
	Cardiac insufficiency	3	2.75
	Liver insufficiency	1	0.92

*CYP2D6*: cytochrome P450 2D6; MPEGS: minimally penetrating glaucoma surgery; MIGS: minimally invasive glaucoma surgery.

**Table 2 pharmaceutics-17-00325-t002:** SNP frequencies in the selected cohort regarding *PTGFR* (A), *ADRB2* (B), and *CYP2D6* (C) genes.

	(A)		(B)	(C)				
	***PTGFR* gene**		***ADRB2* gene**	** *CYP2D6 gene* **				
**rs**	**rs3766355**	**rs3753380**	**rs1042714**	**rs16947**	**rs769258**	**rs3892097**	**rs28371725**	**rs5030656**
**Wildtype (WT)**	N = 3 (2.75%)	N = 57 (52.29%)	N = 17 (15.6%)	N = 44 (40.37%)	N = 98 (89.91%)	N = 65 (59.26%)	N = 77 (70.37%)	N = 105 (96.30%)
**Heterozygous (HT)**	N = 23 (21.1%)	N = 44 (40.37%)	N = 38 (34.86%)	N = 50 (45.87%)	N = 10 (9.17%)	N = 40 (37.04%)	N = 28 (25.93%)	N = 4 (3.70%)
**Homozygous (HM)**	N = 83 (76.15%)	N = 8 (7.34%)	N = 54 (49.54%)	N = 15 (13.76%)	N = 1 (0.92%)	N = 4 (3.70%)	N = 4 (3.70%)	N = 0 (0.00%)
	Study cohort (N = 109)							

*PTGFR*: prostaglandin-F2α receptor; *ADRB2*: beta-2 adrenergic receptor; *CYP2D6*: cytochrome P450 2D6; N: total of participants.

**Table 3 pharmaceutics-17-00325-t003:** Intraocular pressure and visual field parameters according to SNP rs3766355 of the *PTGFR* gene in the PGA group.

*PTGFR* rs3766355 PGA Group						
**bIOP ± SD**		**Kruskal–Wallis (*p*)**	**tIOP ± SD**	**Kruskal–Wallis (*p*)**	**vIOP ± SD**	**Kruskal–Wallis (*p*)**
**WT** (N = 6)	24.00 ± 6.13	*p* = 0.934	20.17 ± 7.41	*p* = 0.982	−17.23 ± 16.44	*p* = 0.677
**HT** (N = 23)	23.39 ± 5.21		20.27 ± 4.22		−8.10 ± 24.80	
**HM** (N = 83)	23.80 ± 4.75		20.75 ± 4.75		−11.64 ± 24.43	
**MD ± SD**		**Kruskal–Wallis (*p*)**	**PSD ± SD**	**Kruskal–Wallis (*p*)**	**VFI ± SD**	**Kruskal–Wallis (*p*)**
**WT** (N = 6)	−2.49 ± 1.91	*p* = 0.153	2.57 ± 0.70	*p* = 0.246	95.67 ± 2.66	*p* = 0.152
**HT** (N = 23)	−3.49 ± 6.13		3.91 ± 3.99		93.61 ± 12.73	
**HM** (N = 83)	−4.45 ± 5.99		3.92 ± 3.05		88.77 ± 17.78	

*PTGFR*: prostaglandin-F2α receptor; PGA: prostaglandin analogues; WT: wildtype; HT: heterozygote; HM: homozygote; bIOP: baseline intraocular pressure; tIOP: treated intraocular pressure; vIOP: rate of intraocular pressure variation; MD: medium deviation; PSD: pattern standard deviation; VFI: visual field index; N: total of eyes; p: probability value; SD: standard deviation.

**Table 4 pharmaceutics-17-00325-t004:** Intraocular pressure and visual field parameters according to SNP rs3753380 of the *PTGFR* gene in the PGA group.

*PTGFR* rs3753380 PGA Group						
**bIOP ± SD**		**Kruskal–Wallis (*p*)**	**tIOP ± SD**	**Kruskal–Wallis (*p*)**	**vIOP ± SD**	**Kruskal–Wallis (*p*)**
**WT** (N = 61)	24.91 ± 4.85	*p* = 0.016 *	20.80 ± 4.53	*p* = 0.425	−15.94 ± 22.10	*p* = 0.032 *
**HT** (N = 45)	22.00 ± 4.66		20.58 ± 5.20		−2.99 ± 25.23	
**HM** (N = 6)	23.67 ± 4.50		18.20 ± 1.79		−20.43 ± 20.24	
**MD ± SD**		**Kruskal–Wallis (*p*)**	**PSD ± SD**	**Kruskal–Wallis (*p*)**	**VFI ± SD**	**Kruskal–Wallis (*p*)**
**WT** (N = 61)	−4.58 ± 6.60	*p* = 0.852	3.94 ± 3.38	*p* = 0.875	89.32 ± 18.66	*p* = 0.956
**HT** (N = 45)	−3.72 ± 5.16		3.92 ± 3.29		90.95 ± 13.62	
**HM** (N = 6)	−1.82 ± 1.50		2.30 ± 0.50		97.00 ± 1.10	

*PTGFR*: prostaglandin-F2α receptor; PGA: prostaglandin analogues; WT: wildtype; HT: heterozygote; HM: homozygote; bIOP: baseline intraocular pressure; tIOP: treated intraocular pressure; vIOP: rate of intraocular pressure variation; MD: medium deviation; PSD: pattern standard deviation; VFI: visual field index; N: total of eyes; p: probability value; SD: standard deviation. * *p* < 0.05 (indicating statistical significance).

**Table 5 pharmaceutics-17-00325-t005:** Intraocular pressure and visual field parameters according to SNP rs1042714 of the *ADRB2* gene in the BB group.

*ADRB2* rs1042714 BB Group						
**bIOP ± SD**		**Kruskal–Wallis (*p*)**	**tIOP ± SD**	**Kruskal–Wallis (*p*)**	**vIOP ± SD**	**Kruskal–Wallis (*p*)**
**WT** (N = 14)	26.25 ± 1.66	*p* = 0.001 *	19.08 ± 3.87	*p* = 0.271	−27.00 ± 15.89	*p* = 0.511
**HT** (N = 18)	19.13 ± 3.50		16.27 ± 4.27		−15.70 ± 24.52	
**HM** (N = 27)	24.29 ± 5.38		19.84 ± 5.73		−20.93 ± 21.09	
**MD ± SD**		**Kruskal–Wallis (*p*)**	**PSD ± SD**	**Kruskal–Wallis (*p*)**	**VFI ± SD**	**Kruskal–Wallis (*p*)**
**WT** (N = 14)	−1.13 ± 1.53	*p* = 0.064	2.07 ± 0.72	*p* = 0.058	91.52 ± 23.73	*p* = 0.217
**HT** (N = 18)	−6.93 ± 6.78		5.75 ± 3.99		82.31 ± 17.96	
**HM** (N = 27)	−5.35 ± 8.92		3.72 ± 2.76		89.42 ± 21.17	

*ADRB2*: beta-2 adrenergic receptor; BB: beta-blockers; WT: wildtype; HT: heterozygote; HM: homozygote; bIOP: baseline intraocular pressure; tIOP: treated intraocular pressure; vIOP: rate of intraocular pressure variation; MD: medium deviation; PSD: pattern standard deviation; VFI: visual field index; N: total of eyes; p: probability value; SD: standard deviation. * *p* < 0.05 (indicating statistical significance).

**Table 6 pharmaceutics-17-00325-t006:** Intraocular pressure and visual field parameters according to the *CYP2D6* metabolizer phenotypes in the BB group.

*CYP2D6* Metabolizer Phenotypes in BB Group						
**bIOP ± SD**		**Kruskal–Wallis (*p*)**	**tIOP ± SD**	**Kruskal–Wallis (*p*)**	**vIOP ± SD**	**Kruskal–Wallis (*p*)**
**PM** (N = 2)	27.00 ± 0.00	*p* = 0.064	14.00 ± 1.41	*p* = 0.010 *	−48.15 ± 5.24	*p* = 0.046 *
**IM** (N = 18)	22.00 ± 4.93		18.33 ± 4.51		−13.72 ± 24.49	
**EM** (N = 23)	23.66 ± 3.75		17.43 ± 4.64		−26.80 ± 15.10	
**UM** (N = 5)	28.67 ± 4.62		25.20 ± 3.70		−10.76 ± 8.71	
**MD ± SD**		**Kruskal–Wallis (*p*)**	**PSD ± SD**	**Kruskal–Wallis (*p*)**	**VFI ± SD**	**Kruskal–Wallis (*p*)**
**PM** (N = 2)	−1.44 ± 0.09	*p* = 0.081	1.88 ± 0.24	*p* = 0.178	96.50 ± 0.71	*p* = 0.232
**IM** (N = 18)	−9.74 ± 9.90		5.90 ± 3.86		78.35 ± 23.11	
**EM** (N = 23)	−3.68 ± 5.66		3.54 ± 2.89		87.19 ± 25.63	
**UM** (N = 5)	−0.69 ± 1.18		1.69 ± 0.45		98.80 ± 0.84	

*CYP2D6*: cytochrome P450 2D6; BB: beta-blockers; PM: poor metabolizer; IM: intermediate metabolizer; EM: efficient metabolizer; UM: ultrarapid metabolizer; bIOP: baseline intraocular pressure; tIOP: treated intraocular pressure; vIOP: rate of intraocular pressure variation; MD: medium deviation; PSD: pattern standard deviation; VFI: visual field index; N: total of eyes; p: probability value; SD: standard deviation. * *p* < 0.05 (indicating statistical significance).

## Data Availability

The original contributions presented in this study are included in the article/Appendix A. Further inquiries can be directed to the corresponding author.

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
