# Peer review of "Pharmacogenetic Influences on Individual Responses to Ocular Hypotensive Agents in Glaucoma Patients"

_pharmaceutics, 2025, doi:10.3390/pharmaceutics17030325_

Round 1
Reviewer 1 Report
Comments and Suggestions for Authors First of all, congratulations to the authors for taking up this topic because the interest in the clinical implementation of pharmacogenetics recently has grown rapidly. Pharmacogenetics, which influences both drug pharmacokinetics and pharmacodynamics, aims to optimize clinical outcomes by facilitating the selection of the most appropriate drug and initial dosage for individual patients. The understanding of genetic factors contributing to variable drug responses has expanded significantly, providing robust evidence to support the use of genotype data to guide pharmacological treatments across multiple therapeutic areas. It is anticipated that glaucoma management may also benefit from this technology in the near future and offer more personalized medicine. However, I have some comments regarding the manuscript/ Comments Authors should clarify the inclusion and diagnostic criteria for ocular hypertension and any subtype of glaucoma. There is no clear statement whether only patients with newly diagnosed glaucoma were included in the study? If I understand correctly following line 235, the patients after glaucoma surgery were also involved into…please clarify why? especially if they had already been treated with antiglaucoma drops before surgery?It seems that the authors conducted the study on very heterogeneous group of patients and different treatment regimens without giving detailed information.
There is no information about general/ chronic disease of the participants, and the impact of some gene polymorphism.
According to the EGS guidelines, antiglaucoma drops as PGA, BB or PM are not the first line treatment, please explain why those patients were included in the study? Had the iridotomy or iridoplasty been performed or cataract surgery before?
In line 170 the authors stated: “ under strict monotherapy regimen” but there is no information on how long the patients were treated, in what regimen, and when the intraocular pressure was re-measured to check the response to ocular hypotensive treatment. How patients were selected for a given treatment regimen?
Please clarify IOP measurements, especially how it was measured, how many times, at what time, by whom and when exactly during the day. What kind of tonometer was used and how it was calibrated?
Do the authors make any corrections to the intraocular pressure value based on the pachymetry result?
Authors should present statistical power analysis and sample size calculation for the study.Were Hardy-Weinberg equilibrium and linkage disequilibrium examined for all SNPs and how?
Why not to use the multiple regression analysis to check if specific phenotype is a risk factor for bad or good response to the treatment?
In line 273 the authors stated that: “40 eyes had required glaucoma surgery” it is not clear before or during the study, please explain and write what kind of surgery it had been.
In line 275-277 the authors mentioned about a switch in the therapeutic regimen, please give an explanation. I suggest exclude the patient from the study if the drug group had been changed.
According to Tabel 1 and hypotensive treatment per eye I suggest to present what specific drugs were used to treat patients in the given groups?
In the additional materials, I would like to suggest to correct Table 2 A and 3A, namely to divide it into 3 groups depending on the treatment used, to see if there were any differences between the groups with different antiglaucoma drops ordinated.
Whether the visual field examination was performed at the visit when the intraocular pressure was assessed to check the response to the treatment? It would be nice to see eventual evolution of visual fields parameters after the treatment.
Reviewer 2 Report
Comments and Suggestions for Authors
This is confusing that the abbreviation “PM” refers to both “prostamides” and “poor metabolizer”.
Long paragraph containing many aspects in one which should be divided into may for the easy of reading.
Lines: 20-22: Re-write because there is awkward use of “which” and objectives not clearly stated.
Lines 77-78: As written, a novice reader may conclude that PM “lacks carboxylic acid being neutral” whereas the fact could be just opposite i.e. it is neutral because it does not contain carboxylic acid group. This ambiguity should be clarified.
Line 102: “worse response” should be replaced with more scientifically valid term.
Line 203: Procedure should have been described in another paragraph. The procedure should start with how DNA was extracted and purified from the patient’s blood sample.
Line 218: Change paragraph before “Procedure.”
Lines 228-229: The sentence is not complete without providing information about what were determined using the Thermo Fisher equipment.
Line 283; Table 1 last row: Data about renal, cardiac, and liver insufficiency are provided but which treatment caused these organ dysfunction is not provided.
Lines 512-516: The beginning two sentences are not conclusion, so should be removed.
Reviewer 3 Report
Comments and Suggestions for Authors
This manuscript explores the pharmacogenetic and epigenetic factors influencing the response to ocular hypotensive treatments in glaucoma patients. It evaluates genetic polymorphisms, particularly in the PTGFR, ADRB2, and CYP2D6 genes, and their correlation with intraocular pressure (IOP) response. The study's findings highlight potential pathways for personalized treatment in glaucoma.
1. Terms like "ultra-rapid metabolizer" and "poor metabolizer" are not consistently abbreviated throughout the text.
2. While epigenetics is mentioned in the title, it is not adequately addressed in the manuscript. Expanding this discussion would align better with the title's scope.
3. The visual presentation of data could be improved. Some figures lack clarity and need more descriptive captions.
4. Certain subgroups, such as poor metabolizers (PM) and ultra-rapid metabolizers (UM), have small sample sizes, limiting the statistical power of the conclusions.
- The manuscript does not explore functional assays to validate the influence of genetic polymorphisms on treatment outcomes.
6. Minor grammatical errors and awkward phrasing detract from the manuscript's readability. For instance, "It is noteworthy that..." appears excessively and could be streamlined.
7. Can the authors provide specific examples of how epigenetic mechanisms might influence the expression of genes such as PTGFR or ADRB2 in glaucoma patients?
8. How can the study's findings on monotherapy be extended to patients on combination therapies, which are more common in clinical practice?
9. Would functional studies (e.g., receptor binding assays or enzyme activity tests) help validate the role of identified SNPs in drug response variability?
10. Could the authors include a summary table comparing the clinical outcomes associated with different SNP profiles?
Round 2
Reviewer 1 Report
Comments and Suggestions for Authors
Comments
There is still no information about the washout period.
According to the EGS guidelines, antiglaucoma drops as PGA, BB or PM are not the first line treatment of primary angle closure glaucoma, please explain why those patients were included in the study? Had the iridotomy or iridoplasty been performed or cataract surgery before?
Please clarify IOP measurements, especially, at what time exactly during the day it was measured, keeping in mind the diurnal fluctuations of intraocular pressure.
Reviewer 2 Report
Comments and Suggestions for Authors
You have satisfactorily answered my comments.
Author Response
Thank you for your thorough review and valuable feedback in order to improve the quality of the article. We appreciate your time and insights, and we are glad that our responses have satisfactorily addressed your comments.
Reviewer 3 Report
Comments and Suggestions for Authors
The authors have addressed all the points.
Author Response

(The authors gave the same response as above.)
